# Binding Efficacy and Thermogenic Efficiency of Pungent and Nonpungent Analogs of Capsaicin

**DOI:** 10.3390/molecules23123198

**Published:** 2018-12-04

**Authors:** Padmamalini Baskaran, Kyle Covington, Jane Bennis, Adithya Mohandass, Teresa Lehmann, Baskaran Thyagarajan

**Affiliations:** 1School of Pharmacy, University of Wyoming, Wyoming, Laramie, WY 82071, USA; Padmabaskaran@yahoo.com (P.B.); jbennis@uwyo.edu (J.B.); amohanda@uwyo.edu (A.M.); 2Department of Chemistry, University of Wyoming, Laramie, WY 82071, USA; kcovingt@uwyo.edu (K.C.); tlehmann@uwyo.edu (T.L.)

**Keywords:** capsaicin, capsiate, capsaicin-β-d-glucopyranoside, obesity, hydrogen bond, molecular docking, thermogenesis, heat

## Abstract

(1) Background: Capsaicin, a chief ingredient of natural chili peppers, enhances metabolism and energy expenditure and stimulates the browning of white adipose tissue (WAT) and brown fat activation to counter diet-induced obesity. Although capsaicin and its nonpungent analogs are shown to enhance energy expenditure, their efficiency to bind to and activate their receptor—transient receptor potential vanilloid subfamily 1 (TRPV1)—to mediate thermogenic effects remains unclear. (2) Methods: We analyzed the binding efficiency of capsaicin analogs by molecular docking. We fed wild type mice a normal chow or high fat diet (± 0.01% pungent or nonpungent capsaicin analog) and isolated inguinal WAT to analyze the expression of thermogenic genes and proteins. (3) Results: Capsaicin, but not its nonpungent analogs, efficiently binds to TRPV1, prevents high fat diet-induced weight gain, and upregulates thermogenic protein expression in WAT. Molecular docking studies indicate that capsaicin exhibits the highest binding efficacy to TRPV1 because it has a hydrogen bond that anchors it to TRPV1. Capsiate, which lacks the hydrogen bond, and therefore, does not anchor to TRPV1. (4) Conclusions: Long-term activation of TRPV1 is imminent for the anti-obesity effect of capsaicin. Efforts to decrease the pungency of capsaicin will help in advancing it to mitigate obesity and metabolic dysfunction in humans.

## 1. Introduction

Natural chili peppers contain both pungent capsaicinoids and non-pungent capsinoids [1]. Capsaicin (CAP) is a pungent capsaicinoid [2] and Capsiate is a classic example of a non-pungent capsinoid [3]. Both CAP and Capsiate bind to and activate their receptor, transient receptor potential vanilloid subfamily 1 (TRPV1) [4,5] protein, which is expressed in neuronal and non-neuronal tissues. Although CAP and capsiate activate TRPV1, fundamental differences exist in their lipophilicity and stability [4], which may, therefore, affect their efficacy. Previous research suggests that CAP induces thermogenesis and nociception while Capsiate has been shown to cause thermogenesis but no nociception [6]. Also, differences between CAP and Capsiate on their oral sensitivity and sensory tolerance have been reported [7]. Such differences can be attributed to the structural differences between CAP and Capsiate. Further, Capsiate has been shown to be less potent than CAP in activating TRPV1 at equimolar concentrations [4]. However, their TRPV1 binding efficiency has not been evaluated before.

CAP induces the browning of white adipose tissues (WAT) and enhances brown adipose tissue (BAT) activation in mice by enhancing the expression of thermogenic proteins in the inguinal white adipose tissue (iWAT) and brown adipose tissue (BAT) [8,9,10]. Further, CAP feeding causes an upregulation of the expression and deacetylation activity of the cellular metabolic sensor sirtuin-1 (SiRT-1) in WAT and BAT [8,9,10]. CAP-induced rise in intracellular Ca^2+^ via TRPV1 is suggested to facilitate its thermogenic effect [10]. Also, clinical research has demonstrated the enhancement of energy expenditure in humans by CAP and Capsiate [11]. Although similar studies have been performed with several combinations of capsinoids and capsaicinoids in rodent models, such studies have used divergent doses and concentrations of these compounds [7,11], which make the interpretation of the results difficult and challenging. Published work suggests that Capsiate exerts an anti-obesity effect [7,12], while it is also shown to activate other forms of TRP proteins like TRPA1 [5]. Also, there is a lack of studies demonstrating the effect of long-term feeding of equivalent concentrations or doses of pure CAP, Capsiate, and other non-pungent derivatives in rodents.

Thus, the main aim of this work is to compare the effects of CAP, Capsiate, and CAP-β-d-glucopyranoside (CAP-β-dgluco) on diet-induced obesity in rodent model. This work presents the effect of thirty-two (32) weeks of feeding wild type mice a high fat diet (HFD) containing a defined quantity of pure CAP, Capsiate, or another derivative of CAP, CAP-β-dgluco. We hypothesized that the structural differences among these compounds affect their ability to bind to and activate TRPV1, which in turn alters their efficacy to stimulate the expression of thermogenic proteins. Therefore, we analyzed their binding efficiency to the TRPV1 protein using molecular docking studies. We correlated the binding efficiency with their abilities to activate TRPV1 in vitro and to induce thermogenic gene and protein expression in the inguinal WAT in vivo. Our study presents novel data on significant differences between the binding efficiency and thermogenic efficiency of pungent and nonpungent derivatives of CAP. This study will advance our fundamental knowledge on the structural implications of pungent and nonpungent derivatives of CAP on their pharmacological effects and therapeutic development.

## 2. Results

### 2.1. Pungent CAP and Its Non-Pungent Derivatives

In this study, we evaluated the effect of a pungent (Capsaicin; CAP) a nonpungent Capsiate and a glucopyranoside derivative of CAP [13]. Their structures and chemical classifications are described in Figure 1.

### 2.2. Effect of Pungent and Nonpungent Analogs of CAP on HFD-Induced Body Weight Gain, Energy, and Water Intake and Metabolic Activity

In order to evaluate whether pungent and nonpungent capsaicin analogs inhibit high fat diet (HFD)-induced obesity, we fed mice either a normal chow diet or a HFD (± 0.01% of CAP, Capsiate or CAP-β-dgluco) for 32 weeks. We also measured the weekly energy and water intake in these mice. As indicated in Figure 2A,B, HFD feeding caused body weight gain in mice and both CAP and Capsiate inhibited this. However, the effect of Capsiate was significantly less compared to CAP. CAP-β-dgluco did not prevent body weight gain. However, no difference in energy or water intake was observed among these feeding groups (Figure 2C,D).

Next, we determined the metabolic activity of these mice by measuring their respiratory quotient (respiratory exchange ratio) and heat production (energy utilization) using CLAMS Oxymax metabolic activity cages. As described in Figure 2E,F, CAP enhanced both the RER and energy expenditure in mice and Capsiate slightly but significantly improved energy expenditure. However, CAP-β-dgluco did not reverse the inhibitory effect of HFD on metabolic activity.

### 2.3. Effect of Pungent and Nonpungent Analogs of CAP on Thermogenic Genes and Proteins Expression in the Inguinal WAT of Mice

Since CAP and Capsiate showed a protective effect against diet-induced obesity, we evaluated the mRNA levels and expression of thermogenic proteins in the inguinal WAT isolated from mice that received HFD (±CAP, Capsiate or CAP-β-dgluco) for 32 weeks. HFD suppressed the mRNA levels of TRPV1, SiRT-1, PRDM-16, PGC-1α, BMP8b, UCP-1, and PPARα; CAP reversed this. Surprisingly, Capsiate slightly but significantly elevated UCP-1 mRNA while failed to alter the mRNA levels of other genes of thermogenesis. CAP-β-dgluco had no effect. The results are summarized in Figure 3.

Next, we analyzed the expression of these thermogenic proteins in the inguinal WAT of these mice. As shown in Figure 4A,B, CAP antagonized the inhibitory effect of HFD on the expression of TRPV1, PPARα, BMP8b, UCP-1, SiRT-1, PRDM-16, and PGC-1α. Inguinal WAT of Capsiate-treated groups showed a slight but significant increase in UCP-1 mRNA. Both Capsiate and CAP-β-dgluco had no effect on other thermogenic proteins. The original Western blots for these proteins are given in Appendix A.

Published research suggests that activation of TRPV1 by CAP is important for the thermogenic and beiging effect of CAP. Since we observed a remarkable difference among the pungent and nonpungent CAP derivatives on HFD-induced body weight gain and reversal of HFD-mediated suppression of thermogenic proteins in the inguinal WAT, we evaluated the effect of these compounds on TRPV1 activation in vitro. For this, we stimulated HEK293 cells that stably expressed TRPV1 with either CAP (1 μM), Capsiate (1 or 10 μM), or CAP-β-dgluco (1 or 10 μM). As shown in Figure 5A, CAP (1 μM) caused the highest influx of Ca^2+^ compared to Capsiate (1 or 10 μM) and CAP-β-dgluco (1 or 10 μM). Interestingly, the effect of Capsiate and CAP-β-dgluco on intracellular Ca^2+^ signals was washable (red and green lines and bars in Figure 5A,B) while that of CAP was not (Figure 5C,D). Further, as illustrated in Figure 5E–G, when the cells are stimulated with either Capsiate (10 μM) or CAP-β-dgluco (10 μM) their effects can be reversed by washing and CAP can stimulate a robust Ca^2+^ influx in these cells. However, neither Capsiate nor CAP-β-dgluco could activate the cells that were stimulated by CAP previously (Figure 5H,I). Also, capsazepine (TRPV1 inhibitor; 10 μM; 90 min pretreatment) suppressed the activating effect of CAP, Capsiate and CAP-β-dgluco in these cells (Figure 5J).

The difference among the aforementioned potencies raises an important question on the efficacies of these compounds to bind TRPV1 protein. To evaluate this, we performed molecular docking studies. As previously discussed by Yang et al. [14], inside the binding channel the vanillyl group in capsaicin (head) points downward to the S4-S5 linker at the intracellular lipid–water interface, while the aliphatic tail points upward to the upper S4 segment (all residue numbering here is based on that used in the PDB ID 3J5R structure). This orientation has been denominated “Tail up, Head down”. The results of the Yang et al. study indicate that the aliphatic tail in capsaicin makes nonspecific van der Waals interactions with TRPV1. Additionally, a hydrogen bond between the amide group in capsaicin (neck) and T550 in TRPV1 is suggested to anchor the ligand to the binding channel. Interactions between the capsaicin head and residues in transmembrane segments S5 and S3 are suggested to induce activation of the protein through structural changes [15]. To further evaluate the effect of CAP, Capsiate, and CAP-β-dgluco on TRPV1, an in silico molecular docking study was performed. Figure 6 shows the results of the docking of all three ligands to TRPV1. The three selected ligands have similar chemical structures, and they all adopt the “Tail up, Head down” orientation in the channel in their best (lowest energy) docked positions. Although oriented in similar fashion in the binding channel, the three ligands exhibit differences in terms of their interactions with some amino acids in TRPV1. CAP (Figure 6B) exhibits a hydrogen bond between the NH hydrogen in its amide group and the oxygen atom attached to the beta carbon in residue T550. The vanillyl group in capsaicin was found to interact with residues: L553 (S4 segment) and L515 (S3 segment) through Alkyl and Pi-Alkyl bonds [16,17,18,19,20] and with S512 (S3 segment) through carbon–hydrogen bond [16,17,18,19,20]. Capsiate (Figure 6C) did not display hydrogen bonds that could be used to anchor the neck of the molecule to the binding channel. The vanillyl group in capsiate exhibits the same interactions described above for capsaicin. However, the docking results indicate that this group also displays conventional hydrogen bonds with residues N551 and T550, both part of the S4 segment. CAP-β-dgluco (Figure 6D) does not bind TRPV1 in the binding channel. Its neck exhibits a hydrogen bond with T435 (S1 segment), and its vanillyl-d-glucopyranose moiety interacts with residue F438 (S1 segment) through a Pi-sigma bond [16,17,18,19,20], and C442 (S1 segment) through a conventional hydrogen bond.

## 3. Discussion

Our results correlate the binding efficiency of pungent and nonpungent analogs of CAP with their efficiency to stimulate thermogenic machinery in inguinal WAT. The pungent CAP could bind to and maximally activate TRPV1 in vitro and counter HFD-induced obesity in mice.

### 3.1. Structural Differences Impact Binding Efficacy

The data presented in this work indicates that the structural differences among CAP, Capsiate, and CAP-β-dgluco have significant impact on their binding to TRPV1 protein, activation of the channel, and upregulation of thermogenic protein expression in the inguinal WAT of mice. CAP and Capsiate are structurally very similar with each other and exhibit similar binding efficiencies. The docking results suggest that the hydrogen bond between the CAP neck and T550 anchor this ligand to the binding channel. The interactions of the vanillyl group in CAP could then produce the structural changes in TRPV1 leading to activation. The anchoring of CAP to the binding channel explains why this ligand remains strongly attached to TRPV1 during activation. The common interactions of the vanillyl groups in CAP and Capsiate (L553, L515, and S512) could explain how the latter ligand can activate TRPV1. However, it is possible that the extra interactions displayed by the vanillyl group in Capsiate (N551 and T550) produce structural changes in TRPV1 different than those elicited by CAP, leading to the lower level of activation evinced through the physiologic and thermogenic actions in mice described previously [9,10,11]. Regarding CAP-β-dgluco, the low degree of activation displayed by this ligand can be attributed to its binding outside the binding channel. Also, the bulkiness of the glucopyranoside group could prevent the binding to TRPV1. This is consistent with our observation that CAP-β-dgluco displayed the least potency of TRPV1 activation, and its activating effects are readily reversible by washing.

### 3.2. Pungent CAP Cy of Capsaicin Is Important for Its Anti-Obesity Effect

Our data illustrates that both CAP and Capsiate exhibited an anti-obesity effect in rodents without altering energy intake, while CAP-β-dgluco was ineffective. Also, CAP exhibited a better protective effect on HFD-induced weight gain compared to Capsiate. This can be attributed to the differences between their activation of TRPV1 and upregulation of thermogenic proteins in iWAT. That is, activation of TRPV1 by CAP produces a robust and sustained elevation of intracellular Ca^2+^ compared to Capsiate and CAP-β-dgluco, which can stimulate the thermogenic mechanisms. This is conceivable since Ca^2+^-dependent signaling processes are shown to be necessary for the effect of CAP [9,10]. Consistently, both Capsiate and CAP-β-dgluco showed weaker activation of TRPV1 (Figure 5) and their effects are reversible by washing off the agonist. Moreover, CAP could activate TRPV1 post-Capsiate or CAP-β-dgluco stimulation. This is suggestive of a partial activating effect of these compounds on TRPV1 due to their binding efficacy. Also, it is evident from our docking studies that CAP-β-dgluco binds to and activates TRPV1 poorly, which accounts for its lack of effect on countering HFD-induced obesity. However, it is surprising that such a moderate activating effect of Capsiate was sufficient to inhibit HFD-induced obesity in mice (Figure 2A,B), enhance energy expenditure (Figure 2F) and increase UCP-1 mRNA (Figure 3). However, neither Capsiate nor CAP-β-dgluco significantly countered the inhibitory effect of HFD on TRPV1 expression in inguinal WAT but CAP was able to counter this effect of HFD. This action seems to be selective for CAP.

Even though Capsiate was not as effective as CAP in preventing HFD-induced body weight gain, it enhanced UCP-1 and increased energy expenditure. These effects of Capsiate further confirm a regulatory role of TRPV1 in energy expenditure and UCP-1. Interestingly, Capsiate was also shown to activate TRPA1 [5]; it could exert its effect via TRPA1 in adipose tissues to prevent obesity could be due to its ability to activate TRPA1. Although the role of TRPA1 has been suggested in BAT thermogenic mechanisms [21,22,23], its activation by Capsiate in the browning of WAT and energy expenditure still remains to be explored. Further studies are warranted to address this.

Our study has identified that CAP binds to and activates TRPV1 with the highest efficacy compared to Capsiate and CAP-β-dgluco. However, its pungency may limit its clinical uses in humans to counter obesity. Therefore, it is important to develop strategies to decrease the pungency of CAP by coating the molecule with biodegradable and bio-absorbable polymers, which can reduce the burst release and therefore decrease the pungency and sensitization. Future research should address the effect of such strategies.

Collectively, our study shows a direct correlation between the binding and activation of TRPV1 and thermogenic efficiency of CAP derivatives. Our data emphasize the requirement for the hydrogen bond between the neck of CAP and the T550 of TRPV1 and the proper interactions between the CAP head and TRPV1 to facilitate a robust activation of TRPV1. However, further studies are required to mutate this residue and evaluate the effect of CAP and other derivatives. Nonetheless, our study provides a good model to correlate the binding efficacy to the thermogenic efficiency of TRPV1 agonists.

## 4. Materials and Methods

### 4.1. Feeding Studies

Adult wild type C57BL/6 mice were obtained from The Jackson Laboratory, USA and colonies were maintained by in-house breeding. Mice were housed at ambient temperature (22–23 °C). Mice were in groups of four animals per cage. Proper husbandry care was followed as per the recommendation of the Institutional Animal Care and Use Committee of the University of Wyoming. For feeding study, we followed the scheme shown in Appendix A. Briefly, mice received either a normal chow diet (NCD), high fat diet (HFD; 60% calories from fat; D12492; Research Diets, Inc. New Brunswick, NJ, USA.), HFD + Capsaicin (CAP; 0.01% in total diet), HFD + Capsiate (0.01% in total diet), or HFD + CAP-β-d-glucopyranoside (CAP-β-dgluco; 0.01% in total diet) from week 6 through to week 38. The weekly weight gain and energy (food)/water intake were recorded and the weighing personnel was blinded on the groups of mice that received the respective diet. All in vivo and in vitro data analyses using the mice and tissues obtained from the mice were blinded.

### 4.2. Metabolic Activity Studies

Metabolic activity studies were conducted using the Comprehensive Laboratory Animal Monitoring System (CLAMS™, Columbus Instruments, Columbus, OH, USA [24]). Mice were individually placed in the CLAMS metabolic cages with ad libitum access to food and water. After acclimatization for 24 h, metabolic parameters including the volume of carbon dioxide produced (*V*CO_2_), the volume of oxygen consumed (*V*O_2_), the respiratory exchange ratio (RER = *V*CO_2_/*V*O_2_) and the energy expenditure [25] for 48 h.

### 4.3. Quantitative RT-PCR Measurements

Inguinal white adipose tissues (iWAT) were collected from NCD or HFD (±CAP, Capsiate or CAP-β-dgluco)-fed mice as per protocols published procedure [8], and were used for quantitative RT-PCR experiments [9,10]. Total RNA from iWAT was isolated using Tri-reagent (Sigma, St Louis, MS, USA) according to manufacturer’s instructions and cDNA was synthesized using Quantitect reverse transcription kit (Qiagen, Valencia, CA, USA) using Q5plex PCR system (Qiagen, Valencia, CA, USA). Real-time PCR was performed using Quantitect SYBR green PCR kit on Q5plex system. No template control and a sample without reverse transcriptase were used as negative controls for every gene of interest. *18s* mRNA was used as the reference gene. Amplification was performed using a 20-µL reaction volume according to manufacturer’s instruction. Each experiment was performed in triplicates for statistical analysis. The following qRT-PCR primers (IDT, Coralville, IA, USA) were used for the quantification of genes.
*18s* forward—5’-accgcagctaggaataatgga-3’; reverse—5’-gcctcagttccgaaaacca-3’;*mtrpv1* forward—5’-caacaagaaggggcttacacc-3’; reverse—5’-tctggagaatgtaggccaagac-3’;*pparα* forward—5’-gtaccactacggagttcacgcat-3’; reverse—5’-cgccgaaagaagcccttac-3’;*sirt-1* forward—5’-tcgtggagacatttttaatcagg-3’; reverse—5’-gcttcatgatggcaagtgg-3’;*pgc-1α* forward—5’-agagaggcagaagcagaaagcaat-3’; reverse—5’-attctgtccgcgttgtgtcagg-3’;*ucp-1* forward—5’-cgactcagtccaagagtacttctcttc-3’; reverse—5’-gccggctgagatcttgtttc-3’;*prdm-16* forward—5’—cagcacggtgaagccattc-3’; reverse—5’-gcgtgcatccgcttgtg-3’;*bmp8b* forward—5’—tccaccaaccacgccactat-3’; reverse—5’-cagtaggcacacagcacacct-3’.

### 4.4. Immunoblotting

iWAT were isolated from NCD (±CAP) or HFD (±CAP, Capsiate or CAP-β-dgluco)-fed mice as described earlier [8]. iWAT was washed with chilled PBS and homogenized in a lysis buffer (50 mM Tris pH 7.5, 250 mM sodium chloride, 0.5% NP40, 0.5% sodium deoxycholate, 2 mM EDTA, 0.5 mM dithiothreitol, 1 mM sodium orthovanadate, and protease inhibitor cocktail) and centrifuged at 14,000 rpm for 20 min at 4 °C and the supernatant was collected in a prechilled microcentrifuge tube. It was centrifuged again at 14,000 rpm for 20 min at 4 °C to remove fat floating on the top. The supernatant was collected, aliquoted and flash frozen in liquid nitrogen and stored at −80 °C until use. The protein concentration of whole tissue lysate was then determined using the Bradford method: equal amounts of protein (40 μg) were separated by SDS-PAGE, transferred to nitrocellulose membrane and immunoblotted with specific antibodies against TRPV1, SiRT-1, UCP-1, and GAPDH. Western blots were performed in triplicates and band intensity was calculated using image J software. The source and concentration of antibodies used are given in Table 1.

### 4.5. Molecular Docking Simulation Studies

All calculations were carried out with Discovery Studio 2016 (BIOVIA, San Diego, CA, USA) on an Intel Xeon 5600 series. The cryo-EM structure of capsaicin-bound TRPV1 [15] (Protein Data Bank code 3J5R) was used as a starting point. The protein was prepared with the standard protein-preparation protocol in Discovery Studio 2016. The CAP, Capsiate, and CAP-β-dgluco ligands were also built and prepared with Discovery Studio 2016 protocols. A cavity search was performed with Discovery Studio 2016 protocols on the prepared protein to identify all possible sites that included the capsaicin-binding channel [14] and can accommodate the selected ligands. Only one cavity including both features was found. The prepared ligands were docked in this cavity using flexible docking. The selected ligands were allowed to adopt different orientations in the binding pocket. The protein residues P501-Y511 and K535-Q560 were allowed flexibility to produce different protein conformations to fit the selected ligands. Additionally, the residues Y511, S512, T550, E570, and I573 were allowed flexibility to achieve side-chain refinement in the presence of the ligands. All flexible residues are located on the protein parent chain B. The results of the flexible docking procedure were analyzed based on the ligands orientations and fitting in the CAP-binding channel. Ligand interactions were analyzed for the best of the conformations. The binding energies for these molecules are given in Table 2.

### 4.6. Chemicals and Drugs

All chemicals and reagents were obtained from Sigma, St Louis, MS, USA. Capsiate was obtained from Alamone Laboratory, Jerusalem, Israel. High fat diet (D12492) was obtained from Research Diets, Inc. New Brunswick, NJ, USA. qRT-PCR primers were obtained from IDT, Coralville, IA, USA and antibodies were from Cell signaling Technology, Danvers, MA, USA, and Santa Cruz Biotechnology, Dallas, TX, USA.

### 4.7. Statistical Analyses

All data are expressed as means ± SEM. Comparisons between groups were analyzed using one-way ANOVA and post hoc analyses were done using Tukey’s test. A *p*-value of <0.05 was considered as statistically significant. All analyses were performed using Microcal Origin 6.0 software (OriginLab, Northampton, MA, USA) and figures were generated using the same program and then converted into image files using the software Adobe Photoshop CS5 Extended (version 12.1; www.adobe.com).

## Figures and Tables

**Figure 1 molecules-23-03198-f001:**
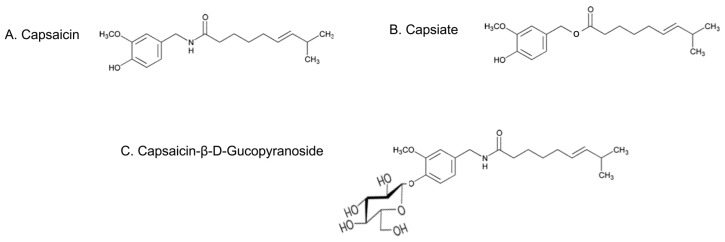
Chemical structures of pungent and nonpungent analogs of CAP. Illustrations of the structures of capsaicin (**A**); (*E*)-*N*-[(4-hydroxy-3-methoxyphenyl) methyl]-8-methylnon-6-enamide; Capsiate (**B**); (*E*)-*N*-[(4-hydroxy-3-methoxyphenyl) methyl]-8-methylnon-6-enoate and Capsaicin-β-d-glucopyranoside (**C**); (*E*)-*N*-[(4-β-d-glucopyranosyloxy)-3-methoxyphenyl) methyl]-8-methylnon-6-enamide.

**Figure 2 molecules-23-03198-f002:**
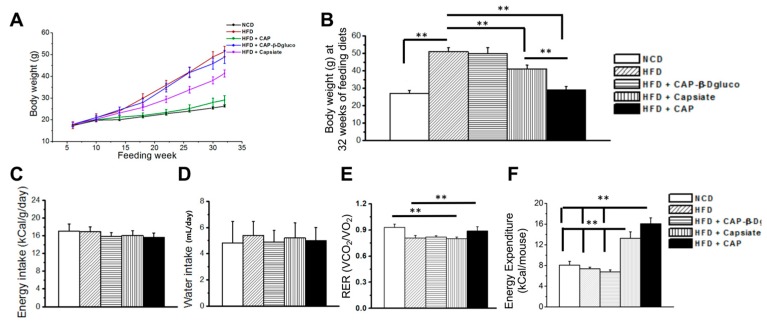
Effect of Capsaicin, Capsiate, and Cap-β-dGluco on body weight gain, energy/water intake, and metabolic activity. **A**. Time course of body weight In NCD or HFD (± CAP, Capsiate or CAP-β-dGluco) in wild type mice plotted against the feeding weeks. **B**. The mean body weight ± SEM for these groups of mice at the end of 32 weeks of feeding. **C** and **D**. The daily mean energy/water intake ± SEM for these mice. **E** and **F**. Mean respiratory quotient (respiratory exchange ratio) and heat (energy expenditure) ± SEM in these mice (*n* = 8 to 12/condition). ** Statistical significance for *p* < 0.01.

**Figure 3 molecules-23-03198-f003:**
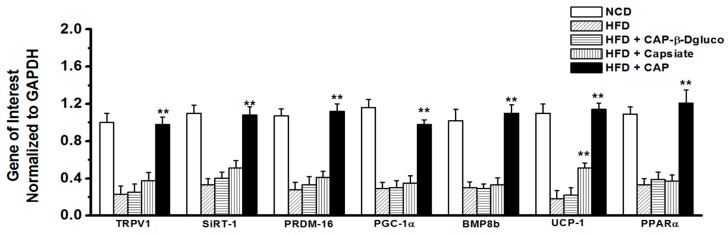
Effect of CAP-β-Dgluco, Capsiate, and CAP on mRNA levels of surrogate thermogenic markers. Mean mRNA levels ± SEM of TRPV1, SiRT-1, PRDM-16, PGC-1α, BMP8b, UCP-1, and PPARα in the inguinal WAT of NCD or HFD (±CAP, Capsiate or CAP-β-dgluco)-fed wild type mice. ** represent statistical significance for *p* < 0.05 for *n* = 4 independent experiments/condition.

**Figure 4 molecules-23-03198-f004:**
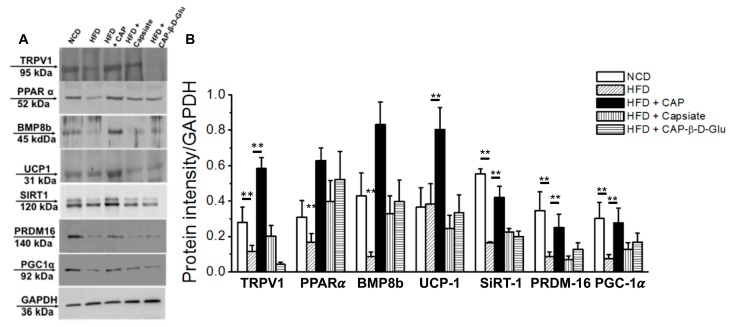
Effect of CAP, Capsiate, and CAP-β-dgluco on thermogenic protein. **A**. Representative western blots showing the expression Of TRPV1, PPARα, BMP8b, UCP-1, SiRT-1, PRDM-16, and PGC-1𝛼 in the inguinal WAT of NCD or HFD (±CAP, Capsiate or CAP-β-dgluco)-fed wild type mice. **B**. Mean protein intensities ± SEM for *n* = 3 experiments/condition. ** represent statistical significance for *p* < 0.05.

**Figure 5 molecules-23-03198-f005:**
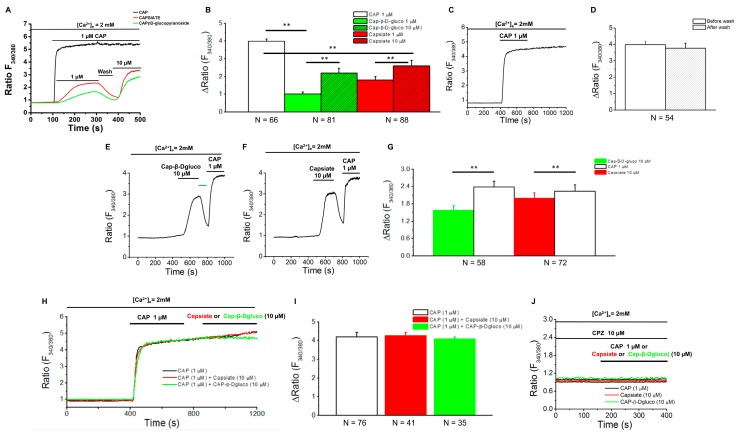
CAP, Capsiate, and CAP-β-dgluco-stimulated Ca^2+^ influx in TRPV1 stably expressing HEK293 cells. **A**. Time courses of intracellular Ca^2+^ influx stimulated by CAP (1 µM), Capsiate (1 and 10 µM) and CAP-β-dgluco (1 and 10 µM). **B**. Mean changes in Fura2-AM fluorescence ± SEM in these cells. **C**. Time course of intracellular Ca^2+^ influx stimulated by CAP (1 µM). **D**. Mean changes in Fura 2-AM fluorescence ± SEM stimulated by CAP before and after wash (with normal extracellular solution) in these cells. **E**. Time course of intracellular Ca^2+^ influx stimulated by CAP-β-dgluco (10 µM) followed by CAP (1 µM). **F**. Time course of intracellular Ca^2+^ influx stimulated by Capsiate (10 µM) followed by CAP (1 µM). **G**. Mean changes in Fura 2-AM fluorescence ± S.E.M in cells treated as per **E** and **F**. **H**. Time courses of CAP (1 μM)-stimulated Ca^2+^ influx in TRPV1 expressing HEK293 cells. Either Capsiate (10 μM; red) or CAP-β-dgluco (10 μM); green) was added at a specific time as indicated in the figure. **I**. Mean fluorescence change ± S.E.M. for cells treated under H. **J**. Time courses of CAP (10 μM), Capsiate (10 μM), or CAP-β-dgluco (10 μM) in Capsazepine (CPZ; TRPV1 inhibitor) preincubated (90 min; 10 μM) cells. CPZ was present throughout the experiment as indicated in the figure. Numbers (N) indicate the number of cells used for obtaining mean ± SEM. ** Statistical significance for *p* < 0.05.

**Figure 6 molecules-23-03198-f006:**
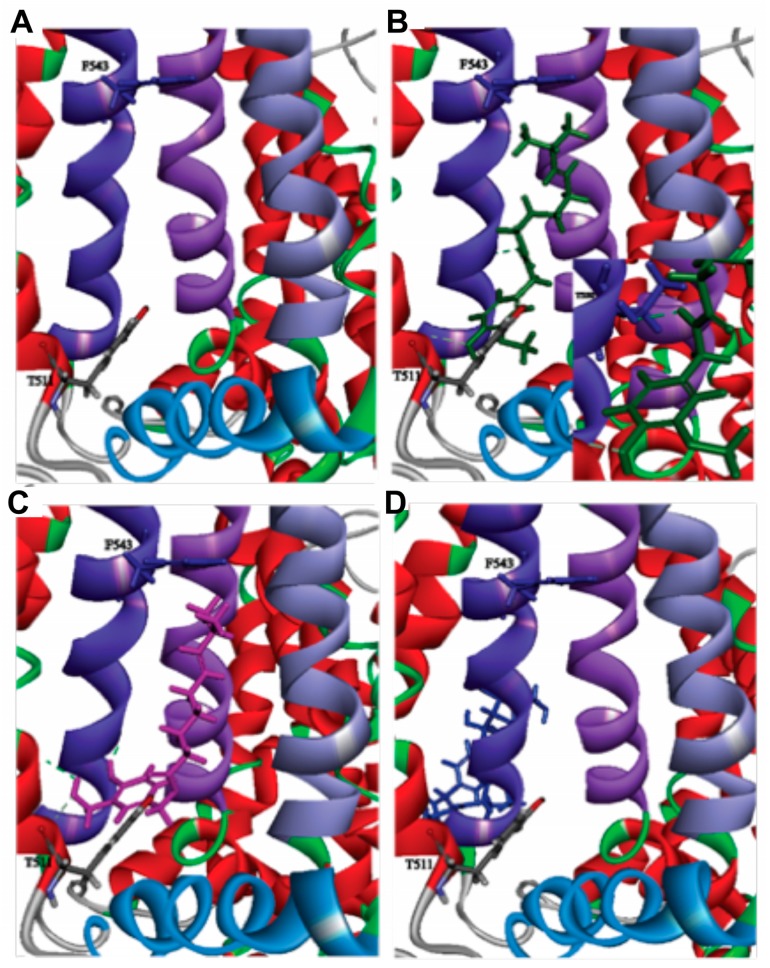
Binding of pungent and nonpungent analogs of CAP with TRPV1. **A**. Empty binding channel in TRPV1. Purple strands left, middle, and right represent the B: S4, D: S5, and D: S6 segments, respectively. The blue strand represents the B: S4-S5 linker. Amino acids F543 and T511 are represented in stick form for reference positions. **B**. CAP bound in the binding channel. The hydrogen bond between CAP and T550 is shown as a green dashed line. **C** and **D.** Capsiate and CAP-β-dgluco bound to TRPV1, respectively.

**Table 1 molecules-23-03198-t001:** Dilutions of antibodies used for immunoblotting experiments.

Antibody	Dilution	Catalog Number and Source of the Antibody
PPARα	(1:500)	NB600-636; Novus Biologicals, Littleton, CO, USA
PRDM-16	(1:1000)	NBPI-77096; Novus Biologicals, Littleton, CO, USA
BMP8b	(1:100)	SC-13086; Santa Cruz Biotechnology, Inc., Dallas, TX, USA
SIRT-1	(1:100)	SC-28766; Santa Cruz Biotechnology, Inc., Dallas, TX, USA
TRPV1	(1:100)	SC-28759; Santa Cruz Biotechnology, Inc., Dallas, TX, USA
PGC-1α	(1:1000)	NBP1-04676; Novus Biologicals, Littleton, CO, USA
GAPDH	(1:500)	SC-365062; Santa Cruz Biotechnology, Inc., Dallas, TX, USA

**Table 2 molecules-23-03198-t002:** Docking energies for CAP, Capsiate, and CAP-β-dgluco.

Ligand Name	Docking Energy
Capsaicin	−27.0486
Capsiate	−27.9297
Capsaicin-β-d-glucopyranoside	+10.0103

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
