# Peer review of "Binding Efficacy and Thermogenic Efficiency of Pungent and Nonpungent Analogs of Capsaicin"

_molecules, 2018, doi:10.3390/molecules23123198_

Reviewer 1 Report

This is an interesting article on plant products.

I recommend publication after some minor changes.

1. The name of the compound 1 can be: (E)-N-(4-hydroxy-3-methoxybenzyl)-8-methylnon-6-enamide

Hence the other compounds can be named likewise.

2. Lines 64, 286, 287 'in vivo', 'in vitro' should be italicized.

Author Response

Responses to the specific comments of Reviewer 1

We thank the reviewer for the kind comments. We have provided the responses to the individual comments of the reviewers below.

Comment 1. The name of the compound 1 can be: (E)-N-(4-hydroxy-3-methoxybenzyl)-8-methylnon-6-enamide

Hence the other compounds can be named likewise.

Response: We have included this in the revised version.

Comment 2. Lines 64, 286, 287 'in vivo', 'in vitro' should be italicized.

Response: We have made this change in the revised version.

Reviewer 2 Report

The manuscript by Baskaran nicely shows that derivatives of capsaicin have different abilities to bind and activate TRPV1, which leads to differential effects on adipose tissue biology and obesity in mice. This manuscript is well written, and the experiments are sound. I have a couple of minor points the authors may want to address:

Minor points:

1. The authors neglect the fact that under most circumstances classical brown adipose tissue (BAT) is the major contributor to non-shivering thermogenesis and its metabolic beneficial effects in mice. What were the effects of the compounds in BAT of these mice here? I am also curious, there was a recent master regulator of brown and beige adipocytes described, does capsaicin have an effect on Nfe2l1 gene expression in BAT or iWAT (Bartelt et al. Nature Medicine 2018)?

2. Figure 2: Lettering in the figures needs to be increased and matched for better visibility. The indication for statistical significance is incorrect to my opinion. Heat/energy expenditure must be calculated per mouse or per lean mass, not per kg body weight (see Tschöp et al. Nature Methods 2012). Then the picture will look different. It is clear that the CAP control displayed strongly increased heat/energy expenditure.

3. Figure 3: Thermogenic surrogate markers better than thermogenic genes.

4. Figure 4: It is unclear how many times the blots were done? Were these biological or technical replicates? The uncropped blots should be annotated and added as supplemental material.

5. Figure 5: Lettering in the figures needs to be increased and matched for better visibility.

Author Response

Responses to the comments of Reviewer 2

We thank the reviewer for the kind comments. We have provided the responses to the individual comments of the reviewers below.

Reviewer 2:

The manuscript by Baskaran nicely shows that derivatives of capsaicin have different abilities to bind and activate TRPV1, which leads to differential effects on adipose tissue biology and obesity in mice. This manuscript is well written, and the experiments are sound. I have a couple of minor points the authors may want to address:  

Minor points:

We thank the reviewer for the kind comments. We have provided the responses to the individual comments of the reviewers below.

Comment 1. The authors neglect the fact that under most circumstances classical brown adipose tissue (BAT) is the major contributor to non-shivering thermogenesis and its metabolic beneficial effects in mice. What were the effects of the compounds in BAT of these mice here? I am also curious, there was a recent master regulator of brown and beige adipocytes described, does capsaicin have an effect on Nfe2l1 gene expression in BAT or iWAT (Bartelt et al. Nature Medicine 2018)?

Response: Thanks for these excellent suggestions. We measured mRNA levels and analyzed protein expression mainly in the inguinal white adipose tissue of mice for this project. This is because we think the browning of white fat by capsaicin feeding is a novel mechanism. We have previously shown the effect of capsaicin on BAT activation and stimulation of thermogenic pathways in BAT (Baskaran et al. Int. J. Obes. 2017). Also, we measured the mRNA levels of NRF2 in the BAT following capsaicin treatment. We found that NRF-2 mRNA levels were decreased in the BAT of high fat diet-fed mice and capsaicin reversed this. We did not include that data here since these experiments have not been done yet for BAT isolated from capsiate and CAP-b-Dgluco fed mice.

 Comment 2. Figure 2: Lettering in the figures needs to be increased and matched for better visibility. The indication for statistical significance is incorrect to my opinion. Heat/energy expenditure must be calculated per mouse or per lean mass, not per kg body weight (see Tschöp et al. Nature Methods 2012). Then the picture will look different. It is clear that the CAP control displayed strongly increased heat/energy expenditure.

Response: We have revised the figure as suggested and have represented the data as kCal/mouse.

Comment 3. Figure 3: Thermogenic surrogate markers better than thermogenic genes.

Response: We have changed this in the revised version

Comment 4. Figure 4: It is unclear how many times the blots were done? Were these biological or technical replicates? The uncropped blots should be annotated and added as supplemental material.

Response: We did the blots thrice from different biological samples. We have included the original uncropped blots as a supplemental data in this revised version.

Comment 5. Figure 5: Lettering in the figures needs to be increased and matched for better visibility.

Response: We have increased the size/font of letters in the figures as suggested.
